# Conceptualizing Core Aspects on Circular Economy in Cities

**Elena Simina Lakatos** [1,2], **Geng Yong** [3,4], **Andrea Szilagyi** [1,2], **Dan Sorin Clinci** [1], **Lucian Georgescu** [5], **Catalina Iticescu** [5,*] **and Lucian-Ionel Cioca** [6,7]

1. Institute for Research in Circular Economy and Environment "Ernest Lupan", 400609 Cluj-Napoca, Romania; simina.lakatos@ircem.ro (E.S.L.); andrea.szilagyi@ircem.ro (A.S.); dan.clinci@ircem.ro (D.S.C.)
2. Faculty of Machine Building, Technical University of Cluj-Napoca, 400641 Cluj-Napoca, Romania
3. School of International and Public Affairs, Shanghai Jiao Tong University, Shanghai 200030, China; ygeng@sjtu.edu.cn
4. School of Environmental Science and Engineering, Shanghai Jiao Tong University, Shanghai 200240, China
5. Faculty of Sciences and Environment, "Dunărea de Jos" University of Galati, 800201 Galati, Romania; lucian.georgescu@ugal.ro
6. Faculty of Engineering, Lucian Blaga University of Sibiu, 10 Victoriei Blv., 550024 Sibiu, Romania; lucian.cioca@ulbsibiu.ro
7. Academy of Romanian Scientists, 010071 Bucharest, Romania
* Correspondence: Catalina.Iticescu@ugal.ro

**Abstract:** Currently, there are many different interpretations in the literature of what a circular economy is and how it functions. As cities are still facing challenges to become fully sustainable, the need for a comprehensive analysis of how the circular economy can be implemented in urban areas is increasing. This article aims at outlining circular cities by their key characteristics and to further explore and provide a framework for fostering circularity at the city level. In order to achieve this goal, we performed a systematic review and analyzed key papers published in the field of circular economy to determine how circular economy practices form circular cities. We discovered that cities play a focal role in facilitating the transition towards circularity through the closing of the loops, recirculation, technical innovation, policy elaboration and citizens' support. However, city policymakers are still uncertain about how a circular city looks like and what its purpose is, as views are ranging from a strategic ambition to a niche concept of a smart city. Such uncertainty brings challenges, especially in the transition phase that many cities are in at the moment. This further implies that circular economy applied at the urban level still needs effort and innovation to successfully pass the transition phase from the linear economy. Therefore, lastly, we developed a framework model that can be adapted in other cities to facilitate their transition to circular cities.

**Keywords:** circular economy; circular cities; sustainability; urban circular economy





## 1. Introduction

As the world is forced to constantly adapt in the face of global climate change and emerging external and internal threats, it is urgent to take appropriate actions to sustain our planet and change our profligate approach towards natural resources [1]. According to the World Bank, the amount of the world population living in urban zones rose from 14% in 1990 to 54% in 2015, and it will eventually reach 66% by 2050, with a tripled urban footprint of urban settlements by 2030 [2–4]. Moreover, cities contribute to 70% of the global greenhouse gas emissions, although this figure may change depending on how a city develops and functions [5].

This rapid shift toward urbanization brings about various challenges such as transportation jams, the safety of the citizens, pollution of all kinds—air, noise, water—and overall, the challenges of the Anthropocene [6,7]. In order to address these concerns, it is critical for all the cities to adopt a holistic and interdisciplinary approach in their planning, their governance, consumers' behaviors, as well as their partnerships and connections, all

combined with novel economic thinking. Under such circumstances, it would be crucial for city managers to encourage technological innovations and initiate new sustainable economic models. Although disruptive at first, technological shifts are necessary, especially through digitalization and new power sources.

New solutions can tackle core issues in an innovative way, allowing for active involvement and goals for further improvement [8]. The transition towards sustainability does not solely depend on technological advancements and policy making but also the involvement of citizens. Without initiatives coming from the inhabitants of the cities and their support, advancements in technology and regulatory frameworks become inefficient [9].

An alternative way to the current way cities develops and operate is provided by the circular economy (CE). The idea of circular economy finds its roots in environmental and ecological economics, industrial ecology and management and corporate sustainability [10,11]. Starting from 1989, when the concept appeared first in an international publication, more initiatives and implementation at various scales took place; for example, in 1996, Germany implemented a new law, the "Closed Substance Cycle and Waste Management Act," to close the waste cycle. Japan adopted a similar policy at an early stage, whereas China used the concept of circular economy as a development strategy from 2008 [12–15].

Although many aspects of the circular economy were already developed in Europe (e.g., resource efficiency, waste legislation, etc.), the European Commission approved the pathway to implement circular economy only in 2015. Today the concept is well accepted by policy-makers, businesses and academia as a "toolbox" for achieving numerous sustainable development targets, such as the SDGs (especially SDG 6-Clean Water and Sanitation; SDG 7-Affordable and Clean Energy; SDG 8-Decent Word and Economic Growth; SDG 11-Sustainable Cities and Communities; SDG 12-Responsible Production and Consumption) [16], the below 2 °C target of the Paris Agreement or the G7 Alliance for Resource Efficiency [8,17–20]. Yet, the concept is currently under systematic changes and debates and still faces many challenges in terms of definitions, assessments, implementation, monitoring or management [21].

The circular city represents a relatively new concept, and as a result, assumptions about these types of cities are often incorrect and require further explanations to understand their way of functioning. A framework for a circular economy not only reduces the raw materials used within the system but also brings opportunities for sustainable consumption, waste management and innovation in all fields, as well as human development and increased well-being for everyone. However, the nature of circular systems mandates for the collective effort of governments, businesses and consumers likewise. Therefore, a circular economy should be an integrated part of the cities' and regions' development plans for achieving healthy circular ecosystems. Nonetheless, the integration of the circular economy cannot be completed without a critical subject knowledge base that can substantiate every strategic decision for the sustainable development of cities and regions, and this is why it is important to review the scientific literature in this field. Considering these facts, this article aims to investigate how circular cities can exist and provide a context for how they can operate in a more efficient and circular way. Previous works have addressed the circular economy in cities, focusing on specific issues such as frameworks for renewable energy [22], water systems [23] or waste management [24]. However, fewer papers have addressed the creation of an integrated framework for how circular cities can function, especially in terms of resource flow.

Therefore, we want to provide a possible answer for how the circular economy can be used effectively as a future city's ultimate goal and driver and what does a circular city need in order to flourish. To answer these questions, we will first analyze how efficient and sustainable circular cities are currently functioning (from a circular economy context) and then elaborate, based on the results, an integrated framework for circularity in cities.

## 2. Cities in the Circular Economy

### 2.1. A Circular Economy Overview

A circular economy is viewed as a "new engine of green growth worldwide" [20]. The primary concept behind circular practices is the development of systems that go beyond linear "take-make-dispose" economic models and aim for closed-loop usage of materials and energy that sustain the value of resources in the economy [25]. In a circular economy, the economic growth is decoupled from resource use by lowering materials input, maximizing the usability and minimizing the waste generation [15,26–29]. With this design, closing the loops aims at solving the problems of resource scarcity, bio-chemical flows and climate change while having a regenerative and restorative benefit for communities [21,30]. Circularity compared to sustainability is a much newer approach, with a deeper interest in minimizing the system inputs, enhancing and preserving the natural sources, efficiency in the management of finite resources and reducing the overall risks. Circularity is also much radial in its design and can be easily associated with the concept of a "self-sustainable" regenerative city [31].

Nevertheless, the circular economy does not remain at the stage of efficiency (results of actions) but exceeds it, adding efficacy (as compared to the effects produced at a strategic, higher level). The efficacy of the economy naturally generates the balance of resources, the environment and the well-being of the inhabitants of the earth in a long term, representing the basis of sustainable development. The circular economy aims at the controlled integration of degenerative activities (active, extensive by definition) and regenerative activities (reactive, intensive by definition in order to preserve the natural balance and, within it, of society, aiming at sustainable development. In addition, these notions can be used in both a smart and resourceful manner for current and future urban planning. To sum up, the circular economy is an umbrella-type concept [32] that, when it is put into practice, has the effect of minimizing environmental impact and stimulating the economy, including slowing raw material inflows and minimizing waste generation, leading to decoupling economic growth of natural resource consumption.

A circular economy aims to create products based on consumers' demands in an eco-friendly and sustainable fashion. This can be difficult and creates the demand for sustainable innovation. Through this process of innovation, new high-tech jobs and technologies can be created to help a circular economy thrive. Furthermore, the shift toward a circular economy system can empower a community by gaining more independence with regards to raw materials and can reduce environmental stress [8]. A circular economy not only reduces the consumption of raw materials but also creates opportunities for sustainable consumption, waste management and innovation in many fields, as well as community development and increased well-being.

However, the circular economy is far from being a perfect concept, and there is still a need to refine circular principles and improve the way they are implemented through various projects and actions. First, public awareness of the benefits and gains of the circular economy is not high, especially in developing countries. This may also be a result of low ecological literacy, but younger generations are more open to reducing resource consumption, recycling and reuse [28]. Another major challenge for the circular economy is its implementation in less economically rich areas, where investments in new high-tech infrastructure are not a possibility. It remains an open question, whether in the absence of investment in new infrastructure (e.g., renewable energy) to replace incumbent unsustainable systems, cities can become truly circular [33]. As far as resource flows are concerned, the Life Cycle Analysis is one of the current instruments used to measure the flows of resources and their impacts on the environment. Although relevant, Life Cycle Analysis is not complete if it solely looks at the material component; it should also embody parts as energy, water or air, for instance. Another tool currently used is the Eco Label, which emphasizes the compatibility of a product with the environment. This label can be used as an integrated aspect of circularity indicators. The current system of indicators is still underdeveloped and fails to capture the wholeness of a city—the definition of

circularity and circular cities are still not complete; therefore, the monitoring system lacks comprehensibility [8].

### 2.2. Circular Consumption and Production

The first core step in a circular economy is the exploration of a region's natural resources and the extraction and utilization of those resources in a sustainable manner. After available natural resources have been analyzed, mining and processing of raw materials can take place. In the next phase, market research should be conducted in order to evaluate consumers' needs. After consumers' needs have been analyzed, the design process begins. In the design process, innovation occurs as engineers and developers create long-lasting, sustainable products that can meet the consumers' needs. This process can be very daunting as it requires manufacturers to shift their thinking from a linear process to a circular one. They have to identify what sustainable materials are needed to ensure the recyclability of the products and also to assess the eco-friendliest production process that generates the least amount of waste and pollution.

The production process involves using recycled and repurposed materials gathered primarily from local, sustainable natural resources and manufacturing that is done in factories that utilize clean renewable energy. In order to reduce pollution and waste, all stages of one production process must be thought of from a circular and sustainable perspective. Each step must be conducted in such a way that nothing is wasted from the materials used to the energy spent on producing the end product [34].

As these products have been developed from a recyclable and reusable perspective from the beginning, it makes the products easier to be used and repurposed. However, consumers must also change their behavior in how they use products. For example, instead of using a product once and simply throwing it away, they must consider alternative options. Some of these options include gifting their products to others who may need them, repurposing these items or recycling them if they no longer serve any purposes.

There are many different ways that consumers can repurpose their used products. Various ways that these products may be repurposed are as simple as taking an old container and turning it into a gardening pot or using pulverized used rubber tires as turf in a playground to prevent injuries from falling. There are also many organizations and initiatives around the world that collect used products in order to repurpose them as gifts or for profit, furthering the ability of a circular economy to function.

If a product is no longer able to be reused or repurposed, the last consumer option in a circular economy would be to recycle the product. This stage becomes easier again as the products were developed from the beginning to be recyclable. However, much depends on consumers to also act in an eco-friendly manner and for local governments to provide efficient recyclable collection containers and to have efficient recycling centers to properly repurpose the products there.

To sum up, a circular economy requires improved collection and processing of recycled products, investments in infrastructure, sustainable design trends and the optimization of a product's life cycle, among many other aspects. At the same time, it requires sustainable education, consumers encouragement and public participation, ecological market development and the promotion and encouragement of reuse and repurposing. Therefore, a circular economy cannot function on the reliance and choices of any one individual or organization but rather on the actions of the entire community. Otherwise, the circular economy lacks systemic validity and relevance and can be easily discredited as its goals are unachievable [21].

### 2.3. Circular Cities: Fundamental Aspects

As cities are expected to host 66% of the world's population by 2050 [35], decision-makers are taking action in the field of sustainability, and the circular economy is one of the models that is gaining momentum. To date, it is more complicated to understand a circular economy by looking at it in its entirety and more difficult to imagine how to create

a circular economy by observing it from this perspective. A circular economy requires many different stakeholders to work together in order to function efficiently. The circular economy practices can be implemented in every part of a functioning economy, at micro-, meso- and macro-levels (*micro* refers to processes in a factory, for instance, *meso-* refers to industrial park or city level and *macro-* refers to regional, national or continental level for instance). In this section, we will discuss the meso level, a level in which the circular economy is based on the city's intrinsic characteristics and sectors and not all of those parts can be created and implemented at the same time [8]. This is why it is critical to first understand how a circular economy functions in an urban context.

At the moment, the majority of cities around the world function in a traditional, linear economic model. They focus their economic activities on commercial products, which are developed as single-life, non-reusable with the aim of serving a singular purpose for immediate convenience [36]. Moreover, cities lack smart and efficient energy and water systems, sharing platforms and operate on inefficient data transmission networks [37] resulting in various pollution issues and resource depletion, as well as more financial capital [38]. Additionally, human effort and time are and will be consumed as the transition from traditional to smarter and circular cities requires more effort [39]. However, such a struggle creates opportunities to rethink the current system and make it less vulnerable, more sustainable and competitive in resource efficiency, waste management and production patterns at different levels.

In the context of urban development, the circular economy can ensure competitiveness, autonomy and multisectoral resilience in the face of upcoming economic and environmental challenges of cities and long-term sustainability. This is achieved by preserving, increasing and (re)using the intrinsic and extrinsic value of all resources. The circular economy involves a circular and rational management of all resources (land, water, energy, infrastructure, goods, etc.). Moreover, the implementation of a circular economy can enhance sustainable growth and economic recovery within the planetary boundaries by restoring natural systems, reducing the negative impacts of climate change and maintaining the minimum use of raw materials. Therefore, the circular economy could be an integrated part of cities' and regions' development plans for closing material loops and a hub for healthy circular ecosystems.

The transition towards a circular economy in cities entails large-scale collaboration between all stakeholders involved; this includes industrial parks as producers, consumers, policy makers and citizens. Moreover, circularity cannot be taken out of the modern digitalized environment as urban spaces are hubs not only for physical products and materials but also data coming from daily activities [7]. In order to fully consider these aspects, cities require an undergoing process of redesign and rearrangement in terms of infrastructure, layout and behavioral patterns, each fully designed to each individual needs. Cities can act as a collaborative platform, which, with an adequate design, can map out synergies to progress linearly to a circular economy.

China and Japan were the first to implement the concept of the circular economy, whereas the model has been applied ever since also in European cities. Examples of the Japanese model have been used in industrial ports such as Dunkerque (France) and Kalundborg (Denmark) [31]. Various circular city experimentations were also conducted in Almere, Amsterdam, Birmingham, Dusseldorf, Genoa, Ghent, Ljubljana, London, Utrecht, etc. through waste management systems, local food systems, industrial symbiosis experiments, material recycling or various strategic plans for circular economy businesses and institutions [31]. At the moment, many cities have already embraced the circular economy concepts and have developed their circular strategy. Through projects, such as EU Horizon 2020 R2pi projects (2016–2019), centred on circular business models and Horizon 2020 CLIC projects (2017–2020) focused on cultural heritage and landscape regeneration and adaptive reuse as drivers of circular economy, more cities are encouraged to become future-oriented, circular cities [31].

As far as production and consumption are concerned, different from a linear economy that focuses on taking precious natural resources to create non-renewable products, a circular economy at the city level functions on recycling and eliminating waste as much as possible. For example, both urban symbiosis and industrial symbiosis are key activities for CE success in cities as they are based on the synergistic opportunity arising from geographic proximity through the transfer of physical resources [40]. These tools are more important as the industry must rethink its profligate approach to resources [16]. The geographical proximity within the urban contexts can facilitate byproducts exchanges among different stakeholders so that the overall wastes can be minimized [40]. This would also facilitate sharing of labor, capital and infrastructure, as well as opportunities for efficient transportation systems and technological spill-overs. Additionally, the minimal and efficient use of raw materials, resource allocation, domestic competitiveness and equal distribution would improve the environmental quality and the overall well-being of the citizens [14,41].

Introducing waste as a resource in the system makes smart recycling both cost-effective and robust. One of the main advantages is that $CO_2$ emission is substantially reduced as a result of fossil fuel resource substitution. The reduced amount of organic solid waste used as fuel compared to the original fossil resources used makes the recycling system more robust because it is not influenced by the changes both in the amount of waste generated and the demand for recycled wastes. Additionally, if a significant amount of waste has the potential to be recycled, the number of incinerators for conventional waste treatment can be reduced [42].

The terms circular and smart cities are often used interchangeably in practice, but this is not the case. Smart cities rely on technology and do not always utilize resources in a sustainable fashion as circular cities do. In a smart city, far more attention is normally placed on how to develop smart systems for a city to operate in a more efficient way. However, it is not necessary for smart cities to function in a sustainable way to still be considered 'smart.' The focus is placed rather on smart design of energy systems, water systems, public transport, waste management, health care, education and infrastructure via cutting edge-technologies and high-speed data transmission. In a smart city, IoT (the internet of things) is fully utilized in order to connect all of the working parts of a city. In general, smart cities can operate in an efficient manner via technological innovation but are not required to function in a sustainable way by definition. However, they can operate in both an efficient and renewable way by incorporating the core principles of a circular economy. By utilizing the core concepts of a circular economy, these cities can function in a way that operates efficiently

To sum up, the circular economy principles at the city level can create an important opportunity to reduce urban waste production and resource consumption towards closed-loop systems [43]. Urban areas represent a fertile environment for implementing, demonstrating and replicating innovative circular solutions as they present a high concentration of resources, capital, data and talent over a small geographic territory [44]. In many ways, a circular city is not a new idea and still requires much effort to exist, and it is not possible to create a circular city in one day. However, the scope of the circular economy has improved from the original concept, as it has assimilated a broader efficiency orientation aside from waste and resources; therefore, the circular economy now also embeds land management, soil protection and even public procurements [14]. The focus needs to be applied to each of the different aspects of a circular economy first in order to create a fully functioning circular city over time.

## 3. Research Method

This study aims at defining circular cities and outlining their core aspects by means of content analysis of the most influential publications in the field. In order to achieve this target, we analyze key papers in the field of circular economy (CE). The goal is to examine how circular economy practices shape circular cities and to propose a framework

model that can be adapted in other cities to facilitate their transition to circular cities. The following sections detail research methods.

We have performed a systematic review according to a predefined search strategy respecting the process stages as suggested by Kitchenham (2004) [45]: (1) planning—justification and protocol; (2) conducting the review—identification of research, selection of primary studies, study quality assessment, data extraction and monitoring and data synthesis; and the final stage; (3) reporting the review results. The selection procedure is similar to that followed by Mainela, Puhakka and Servais (2015) [46] in their review and research agenda pertaining to the concept of international opportunity in entrepreneurship. We propose the following protocol for relevant literature identification (Figure 1):

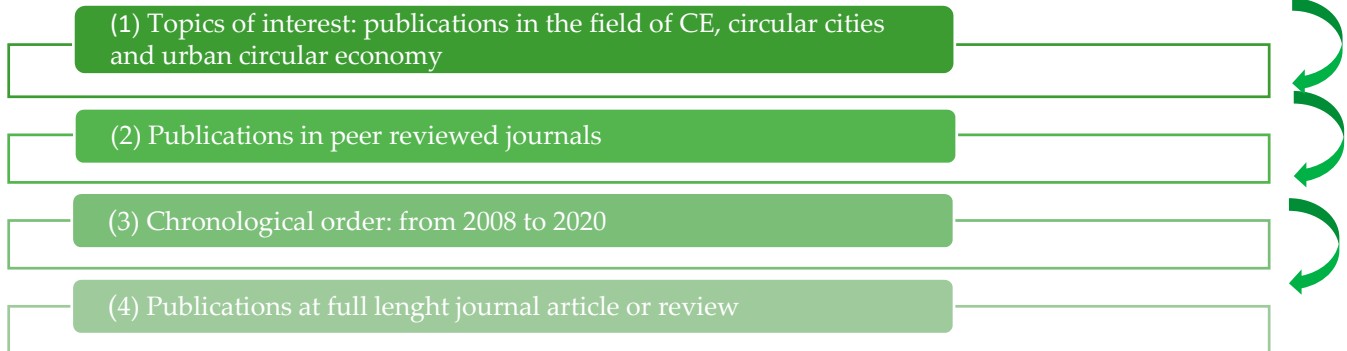

**Figure 1.** Protocol for SE literatures identification.

In the early phase, all the relevant circular economy papers published between 2010 and 2020 have been identified in the ISI Web of Science database. We searched for the topics: "circular economy," "urban circular economy" and "circular cities" in the Web of Science, we analyzed results according to publication years, and we taking into account the characteristics presented in Table 1.

A number of 264 papers were identified, and most of them have been published in journals of Environmental Science, environmental studies and Environmental Management. The first selection was made based on the content of abstracts, the representativeness of which was also weighed on the basis of the authors' names (by excluding papers with similar content) and geographical area. In so doing, 155 most representative articles were selected and grouped according to the different topics of interest for our review. Regarding the selection process, we want to add that the following selection criteria were used: (i) papers published between 2008 and 2020; (ii) articles that present/analyze/explore at least one concept or one study case of circular economy applied in urban settings; (iii) articles that discuss, analyze or propose future directions for the development of circular cities. Review articles, non-indexed studies, book chapters and other reports were excluded. In addition, studies without full text or duplicate articles were excluded as well.

Secondly, we have selected only such materials published in journals with an ISI-impact factor of over 1.00 in the 2020 ranking so as to ensure the quality of materials. Only 53 papers provide a clear understanding of the circular economy and circular cities within their contents (Appendix A). Thus, these papers were selected for further analysis. Moreover, we have selected only research articles rather than books and book chapters, as a wider literature area would affect the quality of the content analysis.

For the content analysis we propose three types of coding: (1) open coding, (2) axial coding and (3) selective coding to construct our theories. Firstly, we have identified and sorted the relevant literature in the field. Secondly, we have extracted innovation-related paragraphs from the materials to be coded according to the typical methodology. Finally, we have formulated a research agenda for the future. The analysis was developed using the content analysis procedure that belongs in grounded theory [55]. As noted, we

have analyzed our materials using three types of coding: open coding, axial coding and selective coding.

**Table 1.** Main limits and challenges of the transition to circular cities.

| Principles of CE | Limits or Challenges at the City Level | References |
|---|---|---|
| Design | Closing the loop of product life cycle through greater recycling and reuse. | Renzulli et al., 2016 [47] |
| Reduction Reuse | Implemented policies fail to bring significant reductions in resource consumption as they only address the efficiency of use but do nothing to reduce the demand for resources. Increasing the consumer demand towards the reuse of products and materials | Kalmykova, Rosado and Patricio, 2016, Lee, Quinn and Rogers, 2016, Prendevrile and Cherim, 2018. [33,48,49] |
| Recycle | Low compliance by the users for the application of the selective collection. Reinforcement of local markets of recycled materials. | Ferronato, Ragazzi, Portillo, Lizarazu and Torretta, 2019, Joensuu et al., 2020. [50,51] |
| Renewable Energy | In cities that have been built without considering the endogenous energy option, the buildings do not incorporate passive or active energy systems. Creating efficient management of energy and materials for the development of urban infrastructure that reduces the constant requirement of these inputs | Baragan-Escandon and Terrados, 2016, Barragán-Escandón, Terrados-Cepeda and Zalamea-León, 2017. [22,52] |
| Sustainable consumption and production | Cities are facing scarce resources and insufficient infrastructure capacity, which require innovations in consumption and production systems to improve quality of life for all. | Cohen and Munoz, 2016. [38] |
| Ecological public procurement | The lack of experience and information among public procurement authorities. A dominant emphasis on price rather than quality. The lack of interaction with markets and a lack of competence among procuring organizations. | Alhola, Ryding, Salmenpera and Busch, 2019, Georghiou et al. 2013. [53,54] |

Open coding allowed us to break the data into pieces and label all relevant data regarding our subject. We have discovered codes describing specific phenomena (properties) that were gathered under a category. The next step has been axial coding, aimed at refining and differentiating categories that had resulted from the open coding [56]. The methodology used was originally proposed by Strauss and Corbin (1990) as it allowed us to discover and establish structures and relationships between all types of data labels (phenomena, concepts and categories).

We continued our coding procedure with selective coding. Selective coding represents a prosecution of axial coding at a higher level of abstraction. At this point, the development and integration of axial coding are compared to other groups that focus on possible concepts or variables of interest. It might imply a search for further examples and evidence for the core categories that will help with the development of case stories [56]. Finally, after completing this process, we divided the relevant articles into three general categories. (i) definitions and core aspects, (ii) exploratory studies and (iii) case studies, as illustrated in Figure 2.

Definition and core aspects

Joensuu, Edelman & Saari, 2020, Zhang et al., 2008, Gretzel et al., 2015, Hollands, 2015, de Jong et al., 2015, Angrisano, et al., 2016, Barragan-Escandon, Terrados-Cepeda, & Zalamea-Leon, 2017, Song et al., 2017, Esmaeilian et al., 2018, Hens, et al., 2018 Marin & De Meulder, 2018, Prendeville & Cherim, 2018, Alhola et al., 2019, Mendoza et al., 2019, Nizetic et al., 2019, Saeumel, Reddy, & Wachtel, 2019, Willam, 2019.
[33,39,51,22,53,57–67]

Exploratory studies

Hanssens, Derudder, & Witlox, 2013, Savini, 2020, Wei, et al., 2014, Zanella et al., 2014, Chen, Gu, Cassidy, & Daganzo, 2015, Sun, et al., 2017, Boix & Leipold, 2018, Yu, Zhao, Fu, & Li, 2018, Barthel et al., 2019, Ferronato, Ragazzi, Portillo, Lizarazu, & Torretta, 2019, Sodiq et al., 2019. [24,36,50,67–74]

CIRCULAR CITIES

Case studies

Lazarevic, Kautto, & Antikainen, 2020, Heurkens & Dabrowski, 2020, Nika et al., 2020, Obersterg, Arlati & Knieling, 2020, Veenstra, Wang, Fan, & Ru, 2010, Molina-Moreno et al., 2018, Cohen & Munoz, 2016, Ezzat, 2016, Kalmykova, Rosado, & Patricio, 2016, Lee, Quinn, & Rogers, 2016, Renzulli et al., 2016, Yu, de Jong, & Cheng, 2016, Godfrey & Oelofse, 2017, Koop & van Leeuwen, 2017, Ragazzi et al., 2017, Ribic, Voca, & Ilakovac, 2017, Schneider et al, 2017, De Medici, Riganti, & Viola, 2018,, van Leeuwen, de Vries, Koop, & Roest, 2018, Zeller, Towa, Degrez, & Achten, 2018, Ferronato & Torretta, 2019, Gravagnuolo, Angrisano, & Girard, 2019 [23,29,31,38,48,49,75–89]

**Figure 2.** The reviewed studies on circular economy and circular cities. References in Definition and core aspects: [22,33,39, 51,53,57–67]; References in Exploratory studies: [24,36,50,67–74]; References in Case studies: [23,29,31,38,48,49,75–89].

## 4. Results of the Review

The analysis reveals that there are several broad definitions for circular cities. An analysis of the papers published on circular cities (Figure 2) shows the fact that relevant articles on the topic of circular cities have been published in journals in the category of environmental, sustainable and engineering science. The search was carried out on the Web of Science using the keywords: "circular cities," "circular economy" and "urban circular economy" for searching for the topic category. The review covers those journal articles published between 2008 and 2020, as this period of time has shown the highest

ISI-impact-related articles about the circular economy, circular cities and smart cities. Prior to 2008, there were approximately 2–3 related published articles per year, with the most published articles between 2016 and 2020.

Successful implementation of the circular economy agenda requires effort at different levels-micro, meso and macro-level. The focus of this paper is the meso-level as it is focused on cities, although many lessons, practices or indicators from the other levels provide highly relevant information for fostering circularity at the city level.

A major conclusion that can be drawn from the selected papers is that circular cities are based on circular economy fundamentals [33]. Cities can be seen as "a complex, heterotrophic artificial ecosystem in which resources are produced and consumed by a variety of activities, initiated by interdependent actors across multiple sectors and scales" [90]. As a hotspot for economic activity, cities play a focal role in facilitating the transition towards circularity as stakeholders are geographically close and can aid the closing of the loops, recirculation, technical innovation, policy elaboration and citizens' support. However, city policymakers are still uncertain about how a circular city looks like and what its role is; views range from a strategic ambition to a niche concept of a smart city or even a tool to be used towards an (unknown) end. The uncertainty brings challenges, especially in the transition phase that many cities are in at the moment [87].

Another key finding is that circular cities are still facing challenges to be fully sustainable and operated in a complete circular economy. Currently, circular economy practices cannot be further promoted and require both technological and managerial improvement. As some products are still being developed using toxic materials, the toxic matters can still be released through the recycling and repurposing process. Worldwide, cities are facing scarce resources and insufficient infrastructure capacity that obviously require innovations in consumption and production systems to improve the quality of life in an urban setting [38].

In terms of energy systems, a vast majority of cities have been built without considering the endogenous energy option; thus, the buildings do not incorporate passive or active energy systems [52]. Creating efficient management of energy and materials for the development of urban infrastructure that reduces the constant requirement of these inputs remains a challenge for the circular cities of the future [22].

Concerning the resource flows in cities, based on our analysis, we can affirm that the currently implemented policies fail to bring significant reductions in resource consumption as they only address the efficiency of use but do nothing to reduce the demand for resources [48]. Increasing the consumer demand towards the reuse of products and materials remains a major challenge to be addressed [33,49]. Moreover, cities are facing increasingly scarce resources and insufficient infrastructure capacity, and this generates an imperious need for innovations in consumption and production systems in order to improve the quality of life for all [38].

## 5. Recommendations and Further Directions for Circular Cities

Boosting circularity means added circular knowledge at the base of the new/revised urban and regional policies. Such knowledge must advise and transform the actual strategies, plans of action and programs and also the management models. It means also added circular knowledge for the urban and regional authorities which manage or are responsible for key sectors such as energy, food, water, waste, education, health, mobility, urban planning and public procurement. Local authorities may lack the information or capacity to implement CE solutions due to a lack of (i) indicators and targets, (ii) awareness of the actual alternative circular options and economic benefits, and (iii) the existence of skills gaps in the workforce and lack of CE programmes at all levels of education (e.g., in design, engineering, business schools). Without an evaluation framework or support from the industry, CE initiatives cannot succeed in urban areas [18].

Based on previous research and the conclusions drawn from the analyzed articles, we concluded that a holistic approach to the flow of resources in the circular cities of

the future is needed. This approach is based on the idea that, in addition to the cycle of high-quality raw materials (already characterizing the linear economy), there must be a circular economy within the capacity of our planet and in terms of human rights and needs. Starting from the various challenges identified related to the transfer of responsibility when we aim to recover the material value through recycling, new issues arise, such as increasing $CO_2$ emissions or releasing toxic substances into the environment.

Our proposed framework to address this issue is illustrated in Figure 3. As it was stated in the results section, circular cities require innovative systems of production and consumption in order to attenuate the crisis of increasingly scarce non-renewable resources [38]. Circular cities also need innovation for resolving the challenge of creating efficient management systems of energy and materials [22]. Therefore, whether we are talking about energy systems, resource management, municipal waste or circular urban design, innovation remains an essential factor that conditions the improvement of all these pillars. For these reasons, Figure 3 proposes a model that explains how innovation contributes to the implementation of the circular economy in cities. As cities hold a variety of technological and human resources in such a condensed space, they have the potential to become a cradle of innovation for an urban circular economy that is based on the 10 Rs of the circular economy (Refusal, Rethink/Redesign, Reduction, Reuse, Repair, Reconstruction, Remanufacturing, Re-offer, Recycle, Regeneration and Recovery).

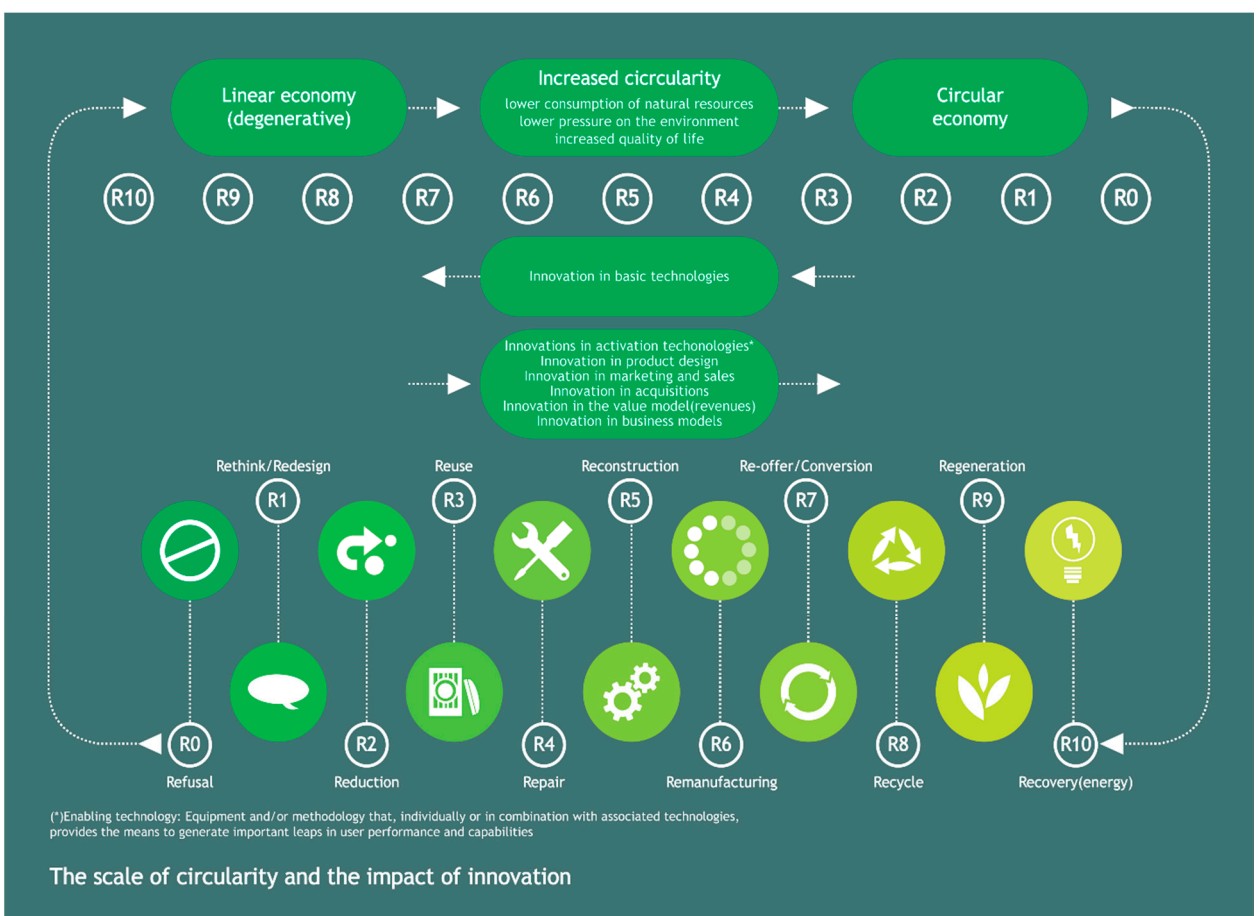

**Figure 3.** The scale of circularity and the impact of innovation in cities.

Figure 4 further presents how innovation, once its presence in a city is facilitated, can enable circular management of materials and resources that take into account the supply circle needs of a city as well as demand cycle needs. The proposed model presents a holistic view of innovative resource management in circular cities, which is based on the principle that in a circular economy, resources can be recovered in a system that is continuous and

long-lasting. Instead of losing their value after the use phase, resources are kept through cycles of reuse, repair, remanufacturing or recycling [91].

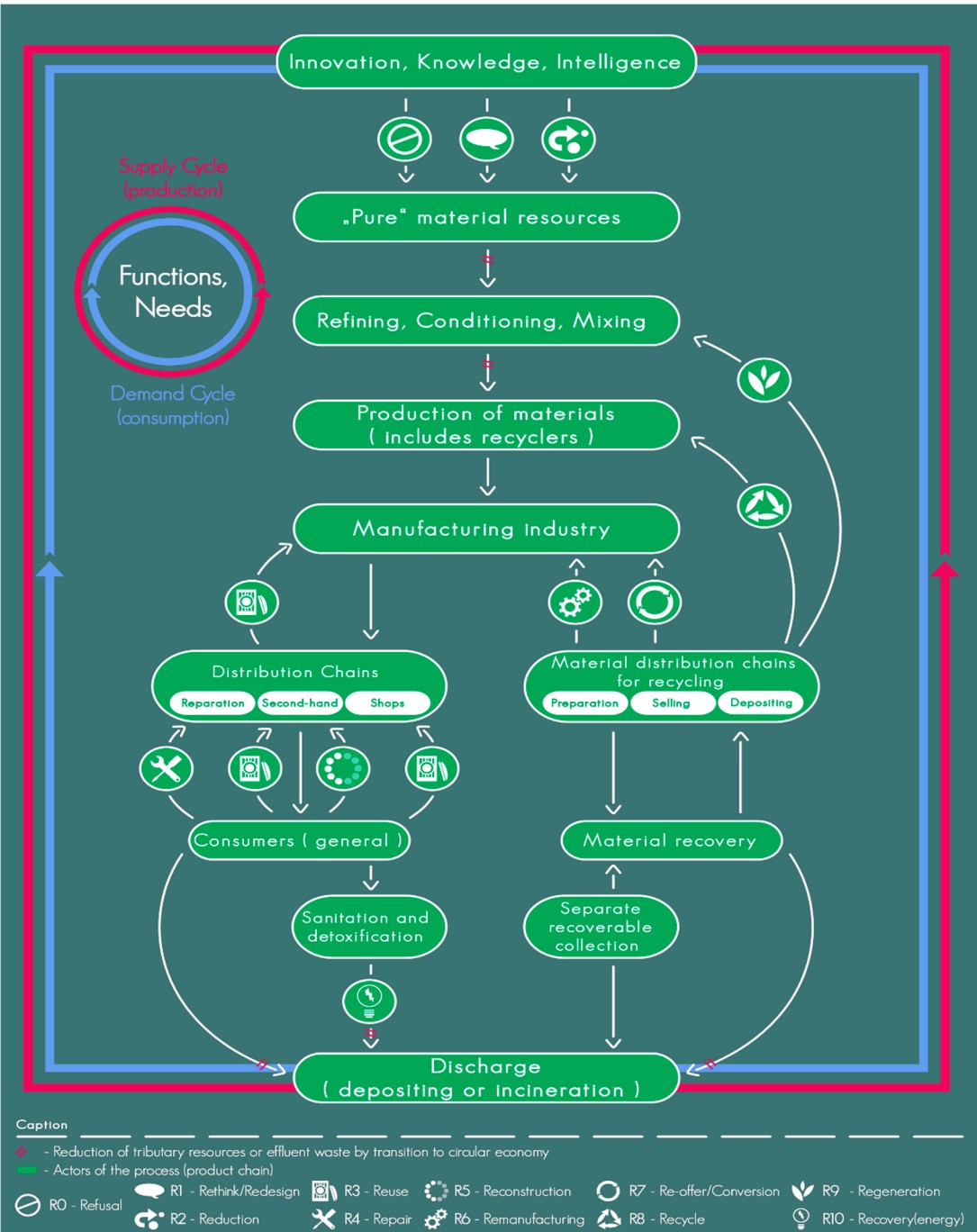

**Figure 4.** The model for resource management circular cities.

The proper functioning of the framework illustrated in Figure 4 at the city level implies the creation of new business models and innovative product designs that use non-toxic materials that can be cycled endlessly. By applying circular principles and strategies, companies can achieve the greatest economic and social values while reducing the negative impact on the environment. Based on this model, the urban circular economy contains both primary circular strategic approaches and secondary circular strategies. Primary circular strategies include prioritizing renewable sources, extending the life of products, using waste as raw material and taking into account new business models. Supporting secondary

circular strategies includes the collaboration of urban agents for value creation, design for the future and integration of digital technologies.

In this way, a circular economy can also create job opportunities in cities—in short, circular jobs, which we can classify as primary employment and employment that supports circular jobs. Together they create direct circular employment opportunities. Additionally, indirect circular jobs are also added, which are not actually circular but still facilitate the circular economy.

## 6. Conclusions

In summary, the circular economy is a key element towards the transition to a regenerative, inclusive and equitable world, whereas, applied at local and regional levels, can enhance the transition to circular urban areas and can address the many systemic crises. Moreover, the implementation of a circular economy can enhance sustainable growth and economic recovery within the planetary boundaries by restoring natural systems, reducing the negative impacts of climate change and maintaining the minimum use of raw materials. A circular economy utilizes natural resources in a sustainable way to develop societies that make use of those resources in a renewable way. An economy that functions in a circular style involves all aspects of an economy, including electricity, water supply, transportation, housing, product development, construction, packaging, agriculture, etc.

This paper contributes from a theoretical point of view to the development of the concept of circular cities primarily by highlighting the current challenges that still remain to be solved, resulted from the analysis of the literature, that are specific to the transition phase to circularity. Secondly, this paper elaborated, based on the results, an integrated framework for circularity in cities and lastly provided suggestions for fostering innovation and circular resource management in cities.

The process to develop a circular economy is a daunting one, and it should be carried out step-by-step rather than attempting to fulfil all of its aspects at once. In reality, it is difficult to create a circular economy as it requires a complete restructuring of current linear systems. However, if attempts are made to focus on taking steps to tackle individual aspects of a circular economy, it is possible to create a fully functioning circular economy. Additionally, cities can provide a fertile environment for implementing, demonstrating and replicating innovative circular solutions. Cities could be uniquely positioned to drive a global transition towards a circular economy because of their high concentration of resources, capital, data and talent over a small geographic territory and could greatly benefit from the outcomes of such a transition [44].

A circular city is a city that functions through the usage of circular economy practices. As previously mentioned, a circular economy is difficult to achieve all at one time, and therefore, a circular city must also attempt to take their restructuring process in stages. For example, a circular city could first focus on creating intelligent and sustainable energy systems and then move on to smart water systems, etc. In addition, these circular cities can also start introducing new sustainable jobs to help build the necessary infrastructure to create a circular city. If a city takes these steps in stride, it will eventually become a fully functioning circular city that functions on intelligent and sustainable usage of natural resources.

Circular cities can benefit and often require smart technologies to function in a sustainable way that fulfills the requirements of a circular economy. Smart technologies can be used to improve resource efficiency to create intelligent electrical, transportation and water systems, etc. [92]. In addition, smart technologies can help to create new jobs and develop innovative products that can be easily recycled or repurposed without generating waste or releasing $CO_2$ or toxins into the atmosphere.

However, we must not forget to mention the limitations of our paper. The literature review we conducted might have missed out on several articles, case studies or exploratory studies that are viable for the scope of our research. Such limitation might be caused by the query construction for our approach, as we selected publications based on the literal use of

the concepts "circular economy," "urban circular economy," and "/" or "circular cities." While using these exact keywords, without a wildcard, we could have missed publications containing terms that are semantically different but with the same meanings.

Based on the results of this paper, we propose several research directions in the field of circular cities. Firstly, future research should take into consideration the fact that, as mentioned in the results section, a lot of policies aimed at circularity fail to achieve a significant reduction in resource consumption since they only tackle efficiency of use but do nothing to reduce the demand for resources. Given this finding, new frameworks that propose new ways to reduce the resource demand of cities could be developed in order to improve the metabolism of the cities in terms of resources intake. Secondly, this paper has found out that innovations in consumption and production systems represent a necessary, topical necessity for cities in the transition to circularity. As cities will continue to face increasingly scarce resources, future papers should propose new theoretical models that focus particularly on sustainable consumption and production models that can improve the quality of life for all in cities. Considering the near future, CE initiatives and actions applied at the urban level can significantly contribute to the transition towards a sustainable, regenerative, inclusive and just circular economy. While it is true that most of the urban and peri-urban areas contribute to resource consumption and waste generation, as well as to global GDP, they are also an important source of innovation and socio-economic transformation.

**Author Contributions:** Conceptualization, E.S.L., G.Y., A.S., D.S.C. and C.I.; methodology, E.S.L., G.Y., A.S., D.S.C. and L.G.; software, E.S.L.; validation, E.S.L., L.G., L.-I.C. and C.I.; formal analysis, E.S.L., G.Y., A.S. and D.S.C.; investigation, E.S.L., G.Y., A.S., D.S.C. and C.I.; writing—original draft preparation, E.S.L., G.Y., A.S. and D.S.C.; writing—review and editing, C.I., L.G. and L.-I.C.; visualization, E.S.L. and L.-I.C.; supervision, E.S.L.; project administration, E.S.L.; funding acquisition, E.S.L. All authors have read and agreed to the published version of the manuscript.

**Funding:** This project is funded by the Institute for Research in Circular Economy and Environment "Ernest Lupan" on research and innovation programmer: GI2018-02, Grant No. 056/23.11.2018.

**Institutional Review Board Statement:** Not applicable.

**Informed Consent Statement:** Not applicable.

**Acknowledgments:** This work was supported by the Erasmus + Programme, SafeEngine project, contract no 2020-1-RO01-KA203-080085.

**Conflicts of Interest:** The authors declare no conflict of interest.

## Appendix A

The list of papers resulted from the content analysis.

| No. | Authors | Title | Journal | Publication Year |
| --- | --- | --- | --- | --- |
| 1 | Joensuu, T., Edelman, H. and Saari, A. | Circular economy practices in the built environment | *Journal of Cleaner Production* | 2020 |
| 2 | Zhang, H., Uwasu, M., Hara, K., Yabar, H., Yamaguchi, Y. and Murayama, T. | Analysis of land use changes and environmental loads during urbanization in China | *Journal of Asian Architecture and Building Engineering* | 2008 |
| 3 | Gretzel, U., Sigala, M., Xiang, Z. and Koo, C. | Smart tourism: foundations and developments | *Electronic Markets* | 2015 |
| 4 | Hollands, R. G. | Critical interventions into the corporate smart city | *Cambridge Journal of Regions, Economy and Society* | 2015 |

| No. | Authors | Title | Journal | Publication Year |
|---|---|---|---|---|
| 5 | de Jong, M., Joss, S., Schraven, D., Zhan, C. and Weijnen, M. | Sustainable-smart-resilient-low carbon-eco-knowledge cities; making sense of a multitude of concepts promoting sustainable urbanization. | *Journal of Cleaner Production* | 2015 |
| 6 | Angrisano, M., Biancamano, P. F., Bosone, M., Carone, P., Daldanise, G., De Rosa, F. | Towards operationalizing UNESCO Recommendations on "Historic Urban Landscape": a position paper | *Aestimum* | 2016 |
| 7 | Barragan-Escandon, A., Terrados-Cepeda, J. and Zalamea-Leon, E. | The Role of Renewable Energy in the Promotion of Circular Urban Metabolism. | *Sustainability* | 2017 |
| 8 | Song, B., Yeo, Z., Kohls, P. and Herrmann, C. | Industrial Symbiosis: Exploring Big-data Approach for Waste Stream Discovery | *24th Cirp Conference On Life Cycle Engineering* | 2017 |
| 9 | Esmaeilian, B., Wang, B., Lewis, K., Duarte, F., Ratti, C. and Behdad, S. | The future of waste management in smart and sustainable cities: A review and concept paper. | *Waste management* | 2018 |
| 10 | Hens, L., Block, C., Cabello-Eras, J. J., Sagastume-Gutierez, A., Garcia-Lorenzo, D., Chamorro, C., et al. | On the evolution of "Cleaner Production" as a concept and a practice | *Journal of Cleaner Production* | 2018 |
| 11 | Marin, J. and De Meulder, B. | Interpreting Circularity. Circular City Representations Concealing Transition Drivers | *Sustainability* | 2018 |
| 12 | Prendeville, S. and Cherim, E. (2018). | Circular Cities: Mapping Six Cities in Transition | *Environmental Innovation and Societal Transitions* | 2018 |
| 13 | Alhola, K., Ryding, S.-O., Salmenpera, H. and Busch, N. J. | Exploiting the Potential of Public Procurement: Opportunities for Circular Economy | *Journal of Industrial Ecology* | 2019 |
| 14 | Mendoza, J. M., Gallego-Schmid, A., Schmidt Rivera, X. C., Rieradevall, J. and Azapagic, A. | Sustainability assessment of home-made solar cookers for use in developed countries. | *Science of The Total Environment* | 2019 |
| 15 | Nizetic, S., Djilali, N., Papadopoulos, A. and Rodrigues, J. J. | Smart technologies for promotion of energy efficiency, utilization of sustainable resources and waste management. | *Journal of Cleaner Production* | 2019 |
| 16 | Saeumel, I., Reddy, S. E. and Wachtel, T. | Edible City Solutions-One Step Further to Foster Social Resilience through Enhanced Socio-Cultural Ecosystem Services in Cities | *Sustainability* | 2019 |
| 17 | Williams, J. | Circular Cities: Challenges to Implementing Looping Actions. | *Sustainability* | 2019 |
| 18 | Hanssens, H., Derudder, B. and Witlox, F. | Are advanced producer services connectors for regional economies? An exploration of the geographies of advanced producer service procurement in Belgium | *Geoforum* | 2013 |
| 19 | Savini, F. | The circular economy of waste: recovery, incineration and urban reuse | *Journal of Environmental Planning and Management* | 2020 |

| No. | Authors | Title | Journal | Publication Year |
|-----|---------|-------|---------|------------------|
| 20 | Wei, Q., Nagi, R., Sadeghi, K., Feng, S., Yan, E., Ki, S. J., et al. | Detection and Spatial Mapping of Mercury Contamination in Water Samples Using a Smart-Phone. | *Acs Nano* | 2014 |
| 21 | Zanella, A., Bui, N., Castellani, A., Vangelista, L. and Zorzi, M. | Internet of Things for Smart Cities | *Ieee Internet of Things Journal* | 2014 |
| 22 | Chen, H., Gu, W., Cassidy, M. J. and Daganzo, C. F. | Optimal transit service atop ring-radial and grid street networks: A continuum approximation design method and comparisons | *Transportation Research Part B-Methodological* | 2015 |
| 23 | Sun, L., Li, H., Dong, L., Fang, K., Ren, J., Geng, Y., et al. | Eco-benefits assessment on urban industrial symbiosis based on material flows analysis and emergy evaluation approach: A case of Liuzhou city, China. | *Resources Conservation and Recycling* | 2017 |
| 24 | Boix, A. P. and Leipold, S. | Circular economy in cities: Reviewing how environmental research aligns with local practices | *Journal of Cleaner Production* | 2018 |
| 25 | Yu, C., de Jong, M. and Cheng, B. | Getting depleted resource-based cities back on their feet again the example of Yichun in China. | *Journal of Cleaner Production* | 2016 |
| 26 | Yu, H., Zhao, Y., Fu, Y. and Li, L. | Spatiotemporal Variance Assessment of Urban Rainstorm Waterlogging Affected by Impervious Surface Expansion: A Case Study of Guangzhou, China | *Sustainability* | 2018 |
| 27 | Colding, J., Barthel, S. and Sörqvist, P. | Wicked problems of smart cities | *Smart Cities* | 2019 |
| 28 | Ferronato, N., Ragazzi, M., Portillo, M. A., Lizarazu, E. G. and Torretta, V. | How to improve recycling rate in developing big cities: An integrated approach for assessing municipal solid waste collection and treatment scenarios | *Environmental Development* | |
| 29 | Sodiq, A., Baloch, A. A., Khan, S. A., Sezer, N., Mahmoud, S., Jama, M., et al. | Towards modern sustainable cities: Review of sustainability principles and trends. | *Journal of Cleaner Production* | 2019 |
| 30 | Lazarevic, D., Kautto, P. and Antikainen, R. | Finland's wood-frame multi-storey construction innovation system: Analysing motors of creative destruction | *Forest Policy and Economics* | 2020 |
| 31 | Heurkens, E. and Dąbrowski, M. | Circling the square: Governance of the circular economy transition in the Amsterdam Metropolitan Area | *European Spatial Research and Policy* | 2020 |
| 32 | Nika, C. E., Vasilaki, V., Expósito, A. and Katsou, E. | Water Cycle and Circular Economy: Developing a Circularity Assessment Framework for Complex Water Systems | *Water Research* | 2020 |
| 33 | Obersteg, A., Arlati, A. and Knieling, J. | Making cities circular: Experiences from the living lab Hamburg-Altona. | *European Spatial Research and Policy* | 2020 |

| No. | Authors | Title | Journal | Publication Year |
|---|---|---|---|---|
| 34 | Veenstra, A., Wang, C., Fan, W. and Ru, Y. | An analysis of E-waste flows in China. | *International Journal of Advanced Manufacturing Technology* | 2010 |
| 35 | Molina-Moreno, V., Nunez-Cacho Utrilla, P., Cortes-Garcia, F. J. and Pena-Garcia, A. | The Use of Led Technology and Biomass to Power Public Lighting in a Local Context: The Case of Baeza (Spain) | *Energies* | 2018 |
| 36 | Cohen, B. and Munoz, P. | Sharing cities and sustainable consumption and production: towards an integrated framework | *Journal of Cleaner Production* | 2016 |
| 37 | Ezzat, A. M. | Sustainable Development of Seaport Cities through Circular Economy: A Comparative Study with Implications to Suez Canal Corridor Project | *European journal of sustainable development* | 2016 |
| 38 | Kalmykova, Y., Rosado, L. and Patricio, J. | Resource consumption drivers and pathways to reduction: economy, policy and lifestyle impact on material flows at the national and urban scale | *Journal of Cleaner Production* | 2016 |
| 39 | Lee, S. E., Quinn, A. D. and Rogers, C. D. | Advancing City Sustainability via Its Systems of Flows: The Urban Metabolism of Birmingham and Its Hinterland. | *Sustainability* | 2016 |
| 40 | Renzulli, P. A., Notarnicola, B., Tassielli, G., Arcese, G. and Di Capua, R. | Life Cycle Assessment of Steel Produced in an Italian Integrated Steel Mill. | *Journal of Cleaner Production* | 2016 |
| 41 | Yu, C., de Jong, M. and Cheng, B. | Getting depleted resource-based cities back on their feet again the example of Yichun in China | *Journal of Cleaner Production* | 2016 |
| 42 | Godfrey, L. and Oelofse, S. | Historical Review of Waste Management and Recycling in South Africa | *Resources-Base* | 2017 |
| 43 | Ragazzi, M., Fedrizzi, S., Rada, E., Ionescu, G., Ciudin, R. and Cioca, L. I. | Experiencing Urban Mining in an Italian Municipality towards a Circular Economy vision. | *International Conference On Technologies and Materials For Renewable Energy, Environment and Sustainability* | 2017 |
| 44 | Ribic, B., Voca, N. and Ilakovac, B. | Concept of sustainable waste management in the city of Zagreb: Towards the implementation of circular economy approach. | *Journal of The Air and Waste Management Association* | 2017 |
| 45 | Schneider, P., Anh, L. H., Wagner, J. and Reichenbach, J. | Solid Waste Management in Ho Chi Minh City, Vietnam: Moving towards a Circular Economy | *Sustainability* | 2017 |
| 46 | De Medici, S., Riganti, P. and Viola, S. | Circular Economy and the Role of Universities in Urban Regeneration: The Case of Ortigia, Syracuse | *Sustainability* | 2018 |
| 47 | van Leeuwen, K., de Vries, E., Koop, S. and Roest, K. | The Energy and Raw Materials Factory: Role and Potential Contribution to the Circular Economy of the Netherlands | *Environmental Management* | 2018 |
| 48 | Ferronato, N. and Torretta, V. | Waste Mismanagement in Developing Countries: A Review of Global Issues | *International Journal of Environmental Research and Public Health* | 2019 |

| No. | Authors | Title | Journal | Publication Year |
|---|---|---|---|---|
| 49 | Gravagnuolo, A., Angrisano, M. and Girard, L. F. | Circular Economy Strategies in Eight Historic Port Cities: Criteria and Indicators Towards a Circular City Assessment Framework. | *Sustainability* | 2019 |
| 50 | Sun, Y., Song, H., Jara, A. J. and Bie, R. | Internet of Things and Big Data Analytics for Smart and Connected Communities | *IEEE ACCESS* | 2016 |
| 51 | Barragán, A. and Terrados, J. | Sustainable cities: An analysis of the contribution made by renewable energy under the umbrella of urban metabolism | *Urban Regeneration and Sustainability* | 2016 |
| 52 | Koop, S. and van Leeuwen, C. | The challenges of water, waste and climate change in cities | *Environment Development and Sustainability* | 2017 |
| 53 | Fujii, M., Fujita, T., Ohnishi, S., Yamaguchi, N., Yong, G. and Park, H. S. | Regional and temporal simulation of a smart recycling system for municipal organic solid wastes. | *Journal of cleaner production* | 2014 |

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
