# Peer review of "Conceptualizing Core Aspects on Circular Economy in Cities"

_sustainability, doi:10.3390/su13147549_

Round 1

Reviewer 1 Report

In general:

  • The paper has an interesting topic and the literature review is interesting and used methodology is appropriate for this type of a study. However, the presentation of results is rather “light” and section 5 seems rather separate from the rest of the study. This paper has potential but it still needs a quite a bit of work.

  1. Introduction
  • Line 52: Why have you highlighted Germany, Japan and China, in particular? Why are these countries particularly relevant?
  • Line 78: The need for this research and the research gap should be justified better. What is the need for this research? How have you defined the research gap? What is your research question?

  1. Theoretical background
  • Line 101: This seems rather brief statement and a bit separate from the rest of the chapter. Please elaborate this or consider moving it to section 2.3.
  • This section presents CE in a rather “roseate” way. How about the challenges related to CE? This section needs also discussion about the critique towards CE and the discovered operational challenges related to CE.
  • I would also challenge the authors to take a more holistic view on CE, considering also the overall critique towards increasing consumption, the role of product design, development and production process in reuse and recycling (instead of highlighting the role of individual consumers) etc.
  • This section (and particularly 2.2 and 2.3 when discussing smart cities and circular cities) needs more references to previous literature. This section should present the “current state of the art”, what is known and what is yet not known about this topic. And with the help of this, this section should convince the reader that there is a need to review the published papers on circular cities and understand more about circular cities.

  1. Research Method
  • Minor remark: (Tuija, Vesa, & Per, 2015) -> These are all first names of the authors. The correct way to cite would be: Mainela, Puhakka & Servais

  1. Results of the review
  • This section is rather “light” and unfortunately fails to present a clear summary of the results. I do not recognize a clear distinction between this section and the theoretical background presented earlier. What did you discover? What did we learn from the systemic literature review?

  1. Recommendations and further directions for circular cities
  • Are these recommendations based on the literature review and its results? If no, why are these recommendations presented in this paper? If yes, how did you draw these conclusions or are these recommendations based on the authors’ own views?
  • Please present the connection between the literature review study and these recommendations more clearly. At present it seems that these recommendations are separate from the literature review and the scientific basis of these recommendations is not convincing.
  • I would also like to see more efforts on highlighting the city context in this section. After all, the main theme of this article was circular cities, not the possibilities and importance of CE in general.

  1. Conclusions
  • This section discusses quite a lot about the general possibilities of CE but too little about the results and mertis of this literature review study. Please present the contribution of this research to both theory and practice. What is the contribution and novelty value of this research?
  • What are the limitations of this study?
  • Based on this research, what are your recommendations for future research in this field?

Author Response

Dear Reviewer,

We very much appreciated the  constructive comments on this manuscript, they have been very thorough and useful in improving this manuscript, so we want to thank you for your time and effort. Our response to your feedback is as follows:

Comments and Suggestions for Authors

In general:

  • The paper has an interesting topic and the literature review is interesting and used methodology is appropriate for this type of a study. However, the presentation of results is rather “light” and section 5 seems rather separate from the rest of the study. This paper has potential but it still needs a quite a bit of work.

Thank your very much your kind observation, we rethought and rewrote the results and discussions part and added the changes in the manuscript.

  1. Introduction
  • Line 52: Why have you highlighted Germany, Japan and China, in particular? Why are these countries particularly relevant?

We highlighted the example of these countries because we considered them examples of good practice in which a country's development programs also take into account the introduction of policies inspired by the circular economy.

  • Line 78: The need for this research and the research gap should be justified better. What is the need for this research? How have you defined the research gap? What is your research question?

Thank you for your suggestion. We detailed this section and formulated our reserch questions and added them in the manuscript (How can the circular economy be used effectively as a future city ultimate goal and driver? What circularity indicators should we take into account when evaluating such cities?)

  1. Theoretical background
  • Line 101: This seems rather brief statement and a bit separate from the rest of the chapter. Please elaborate this or consider moving it to section 2.3.

Thank your for your observation, we moved it to section 2.3

  • This section presents CE in a rather “roseate” way. How about the challenges related to CE? This section needs also discussion about the critique towards CE and the discovered operational challenges related to CE.

We took into account your valuable suggestion and introduced in the manuscript, in section 2.1, lines some critical points related to CE, starting from line 141.

  • I would also challenge the authors to take a more holistic view on CE, considering also the overall critique towards increasing consumption, the role of product design, development and production process in reuse and recycling (instead of highlighting the role of individual consumers) etc.

Thank your for your suggestion, we detailed a more holistic view in section 2.1 of the manuscript.

  • This section (and particularly 2.2 and 2.3 when discussing smart cities and circular cities) needs more references to previous literature. This section should present the “current state of the art”, what is known and what is yet not known about this topic. And with the help of this, this section should convince the reader that there is a need to review the published papers on circular cities and understand more about circular cities.

We tried to point out some new aspect in the manuscript in the sections you mentioned, many thanks for this observation.

  1. Research Method
  • Minor remark: (Tuija, Vesa, & Per, 2015) -> These are all first names of the authors. The correct way to cite would be: Mainela, Puhakka & Servais

Thank you very much, we made the required change.

  1. Results of the review
  • This section is rather “light” and unfortunately fails to present a clear summary of the results. I do not recognize a clear distinction between this section and the theoretical background presented earlier. What did you discover? What did we learn from the systemic literature review?

Thank your for these observations, we tried to change the result section of the manuscript so that the key findings are better highlighted.

  1. Recommendations and further directions for circular cities
  • Are these recommendations based on the literature review and its results? If no, why are these recommendations presented in this paper? If yes, how did you draw these conclusions or are these recommendations based on the authors’ own views?
  • Please present the connection between the literature review study and these recommendations more clearly. At present it seems that these recommendations are separate from the literature review and the scientific basis of these recommendations is not convincing.
  • I would also like to see more efforts on highlighting the city context in this section. After all, the main theme of this article was circular cities, not the possibilities and importance of CE in general.

We took into account all of your observations, and tried our best to add them into the recommendations section of the manuscript.

  1. Conclusions
  • This section discusses quite a lot about the general possibilities of CE but too little about the results and mertis of this literature review study. Please present the contribution of this research to both theory and practice. What is the contribution and novelty value of this research?
  • What are the limitations of this study?
  • Based on this research, what are your recommendations for future research in this field?

We introduced the limitations and future research directions, thank you for mentioning them, and we added detaild about the contribution. 

Reviewer 2 Report

Please include your key findings in the abstract.

Please avoid using terms like environmentally friendly or green (hard to know what they mean). And consider how differently growth/consumption/waste are maybe addressed in this context (CE).

Use e.g. circular economy and sustainability and define what you mean in each context.

Please provide more system level circular economy references.

Please introduce Table 1 in the text.

Please provide clear results section that is directly based on all the elements of the materials and methods section. And clear figures/tables with results (not just articles).

Please provide clear conclusions section and recommendations/future research focus areas after that section.

Please place models and visions ect in the early chapters to provide a more logical structure.

Please justify if you propose novel approaches for CE but your research focuses on articles. State your messages based on your findings.

Please see if you cand find actual CE strategies ect by cities for references and EU level policies ect.

Author Response

Dear Reviewer,

We want to thank you for the time you spent reading this manuscript but also for the constructive suggestions and comments that helped us a lot. Our answers to your feedback is as follows:

Point 3: Comments and Suggestions for Authors

Please include your key findings in the abstract.

We included the key findings, thank you for your observation.

Please avoid using terms like environmentally friendly or green (hard to know what they mean). And consider how differently growth/consumption/waste are maybe addressed in this context (CE). Use e.g. circular economy and sustainability and define what you mean in each context.

We took into account your suggestions and starting from line 108 we made the differentiation between sustainability and circular economy. We also replaced the terms environmentally and green with more suitable terms.

Please provide more system level circular economy references.

Thank your for your suggestion, we made several changes to sections 2.1 and 2.3 that describe system level CE references.

Please introduce Table 1 in the text.

We tried to format the table as close as possible to the text so that it can still be comprehensible.

Please provide clear results section that is directly based on all the elements of the materials and methods section. And clear figures/tables with results (not just articles).

Thank you for your suggestion, we rewrote several paragraphs and we hope that the result are more convincing now.

Please provide clear conclusions section and recommendations/future research focus areas after that section.

We introduced future research directions, as well as limitations in the conclusion section, thank you for this observation.

Please place models and visions ect in the early chapters to provide a more logical structure.

Thank you for this suggestion, we tried our best to delete some parts that were redundant and to add new relevant paragraphs.

Please justify if you propose novel approaches for CE but your research focuses on articles. State your messages based on your findings.

We introduced in the conclusion section more details about the contribution of this paper.

Please see if you cand find actual CE strategies ect by cities for references and EU level policies ect.

Thank your for your suggestion, we introduced some details about this in section 2.3, starting from line 256.

Reviewer 3 Report

Dear Authors,

Thank you for submitting your interesting paper. I had the pleasure to read it. I have however some feedbacks that can improve the quality of the paper:

  1. I think you may explain a bit more about the selection process of the articles that you have done content analysis.
  2. you have mentioned that most journals are from Environmental Science but it could be Environmental studies and/or Management. There are some nuances between these terms.
  3. I really liked the first part of the paper till the research methodology. However, your findings seem the things we generally know. With the exhaustive coding process that you have done, I think you might come up with some more specific concepts/ issues that we don't know in general. Please review your coding process and see if you can come up with some more latent variables/ issues that can be helpful to show the contribution of your work. In current state, it is difficult to see the contribution of your paper.
  4. Please explain the theory building process and how your paper is helpful/contributing in this regard.
  5. Please add the list of papers you have done the content analysis along with the name of the journal where those papers were published. As "Sustainability" journal is flexible in terms of numbers of pages of the article, the list can be helpful for the readers to understand  your study.
  6. Thank you.

Author Response

Dear Reviewer,

Thank your very much for your valuable time, comments and suggestions, we really appreciate your effort. Our response to your feedback is as follows:

Comments and Suggestions for Authors

Dear Authors,

Thank you for submitting your interesting paper. I had the pleasure to read it. I have however some feedbacks that can improve the quality of the paper:

  1. I think you may explain a bit more about the selection process of the articles that you have done content analysis.

Regarding the selection process, we want to add that the following selection criteria were used: (i) papers published between 2008-2020; (ii) articles that present/analyze/explore at least one concept or one study case  of circular economy applied in urban settings (iii) articles that discuss, analyse or propose future directions for the development of circular cities.

Review articles, non-indexed studies, book chapters and other reports  were excluded. In addition, studies without full text or duplicate articles were excluded as well. We added this details.

  1. you have mentioned that most journals are from Environmental Science but it could be Environmental studies and/or Management. There are some nuances between these terms.

Yes, we agree that there some nuances between these terms, thank you for your observation. We added in the manuscript Environmental Studies and Environmental Management  as sources as well.

  1. I really liked the first part of the paper till the research methodology. However, your findings seem the things we generally know. With the exhaustive coding process that you have done, I think you might come up with some more specific concepts/ issues that we don't know in general. Please review your coding process and see if you can come up with some more latent variables/ issues that can be helpful to show the contribution of your work. In current state, it is difficult to see the contribution of your paper.

Thank you for your observation. We rethought and rewrote the results and discussions part considering the observation made by you and we tried to better connect the analysis of the literature to the results in the updated manuscript.

  1. Please explain the theory building process and how your paper is helpful/contributing in this regard.

Regarding the process of building the theory, in the introductory part we wanted to place the current context of cities along with the challenges they face, to make the reader aware of the urgent need for a new model of urban functioning. After the introduction, in section 2.1 “A circular economy overview” we presented this circular economy as a possible solution to the challenges mentioned in the introduction and in section 1.2 we detailed aspects regarding consumption and production, because as it is well known, most of the resource consumption takes place in cities. After these sections, we considered that we can describe the circular cities and some of their fundamental aspects. We hope that we have been able to clearly explain the reasoning of these sections and the theoretical information related to them. Our paper contribution is to create and holistic, integrated model for how cities can operate in a more efficient and circular way and suggestions for a system of indicators for assessing circularity at the city level.

  1. Please add the list of papers you have done the content analysis along with the name of the journal where those papers were published. As "Sustainability" journal is flexible in terms of numbers of pages of the article, the list can be helpful for the readers to understand your study.

Thank you.. 

5- We added the list of papers as a table, starting from the line 360 of the manuscript.

Round 2

Reviewer 1 Report

In general:

Overall, the made revisions have improved the paper. However, the major weakness of the paper remains: section 5 is still rather disconnected from the conducted study and to me does not present new knowledge about circular cities but rather provides nice-looking figures concerning circular economy in general. This section could still be much improved.

  1. Research method:

- Two consecutive tables cut the flow of the paper, especially as table 1 is very long. I recommend moving table 1 to the end of the paper as an appendix.

- Line 390: You refer to “the figure below”. Please refer to the number of the figure.

  1. Results of the Review:

- On line 409 you state that the focus of this paper is the macro-level. However, in the theoretical background section you focused on the meso-level? If your results focus on the macro-level, shouldn't your theoretical background section also focus on this level? Also, what does it mean that “as it is focused on cities which fall into this category”? Which category? Please elaborate.

  1. Recommendations and further directions for circular cities

- Line 485: Highlight the views of this paper based on your research and previous literature, not the authors’ personal thinking.

- Figure 3: This is a nice-looking figure but it still presents a rather general model of circular economy instead of a model of circularity IN CITIES. The context is missing. Also, you only refer to figure 3 after the figure (and after figure 4) is presented (line 525). What is the novelty value of this figure and why is it presented here?

- Figure 4: When explaining the figure to the reader, you state that circular economy has 7 basic principles to which you added additional 4. What are these 7 principles and which are the ones you added? What are your additions based on? Your literature review? How? Again, this figure should be based on your literature review’s results. Also, you state that this figure presents performance indicators of circularity? How? What are these indicators? I do not recognize measurable performance indicators in this figure. If these claimed indicators are one of the key contributions of this paper, maybe they need to be presented in a more clear way for example as a separate table. To me the problem of figure 4 seems to be that you have tried present too many things in one figure and hence failed to present your key contribution clearly enough to the reader.

- Line 540: “the classification of basic circular principles in the previous section”. In section 4? Where in section 4 are these basic principles classified?  

  1. Conclusions

- Based on this research, what are your recommendations for future research in this field? In the end of section 3 you state that: “Finally, we have formulated a research agenda for the future.” But in the end of the paper you do not present any suggestions for future research in this field.

Author Response

Dear Editor and Reviewers,

       Thank you very much for your valuable comments and suggestions concerning our manuscript.  Your feedback is very helpful for revising and improving our paper, thus we have studied your  comments carefully and have made several corrections which we hope meet with approval. The main corrections in the manuscript, marked with ”track changes” function. Our response to the reviewer’s comments are as following:

Research method:

- Two consecutive tables cut the flow of the paper, especially as table 1 is very long. I recommend moving table 1 to the end of the paper as an appendix.

  • Thank you very much for your suggestion, we agree that table 1 is indeed very long, so we moved it at the end of the paper, as an appendix.

- Line 390: You refer to “the figure below”. Please refer to the number of the figure.

  • We are sorry for the incovenience, we refered to the number of the figure in the revised manuscript.

Results of the Review:

- On line 409 you state that the focus of this paper is the macro-level. However, in the theoretical background section you focused on the meso-level? If your results focus on the macro-level, shouldn't your theoretical background section also focus on this level? Also, what does it mean that “as it is focused on cities which fall into this category”? Which category? Please elaborate.

  • Thank you for your important observation. A typing error occurred on line 409, the level we are referring to is meso, the same level that the theoretical part approaches, as you pointed out. We apologise for this error. By ”category”, we wanted to actually refer to the meso level, but we reformulated the text to be clearer and not mislead readers.You can find all the changes from line 408 to 411.

Recommendations and further directions for circular cities

- Line 485: Highlight the views of this paper based on your research and previous literature, not the authors’ personal thinking.

  • We took into account your comments and tried to make the connection with the result more obvious in this version of the manuscript.

- Figure 3: This is a nice-looking figure but it still presents a rather general model of circular economy instead of a model of circularity IN CITIES. The context is missing. Also, you only refer to figure 3 after the figure (and after figure 4) is presented (line 525). What is the novelty value of this figure and why is it presented here?

  • Thank your for these valuable suggestions, we provided the necesarry context for figure 3 and added more details in the manuscript regarding the city level, focusing on circular innovation in cities.

- Figure 4: When explaining the figure to the reader, you state that circular economy has 7 basic principles to which you added additional 4. What are these 7 principles and which are the ones you added? What are your additions based on? Your literature review? How? Again, this figure should be based on your literature review’s results. Also, you state that this figure presents performance indicators of circularity? How? What are these indicators? I do not recognize measurable performance indicators in this figure. If these claimed indicators are one of the key contributions of this paper, maybe they need to be presented in a more clear way for example as a separate table. To me the problem of figure 4 seems to be that you have tried present too many things in one figure and hence failed to present your key contribution clearly enough to the reader.

  • We agree with your observation, we tried to put to many things in one figure, so in order to present the contribution of this figure we renounced to focus on indicators and rather concentrated on resources in cities.

- Line 540: “the classification of basic circular principles in the previous section”. In section 4? Where in section 4 are these basic principles classified?  

  • Thank you again for your observation, we corrected that paragraph. It was an error of the text editing made by us, because in the previous section there is no classification of principles, as you mentioned very well.

Conclusions

- Based on this research, what are your recommendations for future research in this field? In the end of section 3 you state that: “Finally, we have formulated a research agenda for the future.” But in the end of the paper you do not present any suggestions for future research in this field.

  • Starting from line 619 of the manuscript, we introduced future research directions as well, many thanks for your observation.

Reviewer 2 Report

Good working improving the article.

Please check Figure 3 citation in the text (missing?)

Author Response

Dear Reviewer,

Thank your very much for your observation, the citation of figure 3 was misplaced, but we rectified this, and added the citation in the manuscript.

Reviewer 3 Report

Dear authors,

Thank you for submitting the revised version. I am satisfied with the revised version except the abstract. Given that the abstract plays important role in citation, please include implications of your findings. Please consult a "structural abstract" and that can help you to improve your abstract writing. 

Thank you.

MM.

Author Response

Dear Reviewer,

Thank you very much for your suggestion, we agree that the abstract plays an important role in citation, therefore we took your comments into account and added details about the implication of our paper.

Round 3

Reviewer 1 Report

I'd like to thank the authors for their kind responses and for following the advice they have received during the review process. I think they have really been able to improve this manuscript from the first version, and I'm sure this will be an interesting paper also to other researchers and experts working in the circular economy field.